# Patient Engagement in Providing Telehealth SUD IOP Treatment: A Retrospective Cohort Study

**DOI:** 10.3390/healthcare12242554

**Published:** 2024-12-18

**Authors:** Joanna Contreras-Schwartz, Conor O’Neill, Annelise Threlkeld, Erin O’Callaghan, Mirene Winsberg

**Affiliations:** Brightside Health, San Francisco, CA 94131, USAannelise.threlkeld@brightside.com (A.T.); erin@brightside.com (E.O.); mimi.winsberg@brightside.com (M.W.)

**Keywords:** telehealth, intensive outpatient, substance use disorders, engagement, abstinence

## Abstract

Background: Substance use disorders (SUDs) remain a growing public health issue, with drug- and alcohol-related deaths continuing to increase. A myriad of barriers prevent many with SUDs from seeking care. Telehealth interventions are well-positioned to reduce barriers and increase engagement in SUD treatment. The SUD intensive outpatient program (IOP) is specifically designed for telehealth and offers evidenced-based care delivered by SUD professionals as well as asynchronous assignments to enhance treatment. This study explores the feasibility of providing a telehealth IOP. Methods: participant engagement, reasons for disengagement, and days of abstinence were examined using existing records from a cohort of participants between 2021 and 2023 (*n* = 4724). Results: Nearly 80% of participants remained engaged in the program for 30 days, and 91% attained at least 30 consecutive days of abstinence over the course of treatment. Nearly 45% demonstrated a successful response to care and no longer required IOP treatment. Those who finished the IOP completed over 70% of the asynchronous assignments. Conclusions: Results support the feasibility and effectiveness of delivering a telehealth IOP for SUDs.

## 1. Introduction

Substance use and addiction are widespread public health issues, impacting 168.7 million people ages 12 or older in the United States, with 48.7 million meeting the diagnostic criteria for substance use disorders (SUDs) [1]. Further, the CDC reports that the number of alcohol-related deaths have increased by nearly 30% from 138,000 in 2016–2017 to 178,000 in 2020–2021 [2]. Drug overdose deaths also rose by 80%, with 52,902 reported in 2015–2016 and 94,788 in 2020–2021 [3]. Various evidenced-based SUD treatment modalities are available including medical interventions such as medication-assisted treatment (MAT) [4,5,6,7,8] and psychotherapy-based interventions such as cognitive behavioral therapy (CBT) [9,10,11,12,13], motivational interviewing (MI) [14,15,16], mindfulness-based skills (MBS) [16,17,18,19,20], and/or a combination of these treatment approaches [14,15,16]. While individual therapy is typically a component of SUD treatment, group therapy has long been the primary modality used, with various studies underscoring its effectiveness [21,22,23].

The gold standard of SUD treatment has long been based on in-person treatment [24]; however, various barriers including access to local treatment facilities [25,26], readiness to change [27,28], and stigma [26,29] deter many in need of treatment from obtaining it. Of the 48.7 million individuals meeting the criteria for an SUD, only 6% received SUD treatment [1]. Engaging and retaining individuals with SUDs in treatment presents a challenge, with disengagement rates ranging from 17 to 57% [30] across levels of care, and disengagement from intensive outpatient programs (IOPs) varying from 32% [31] to 50% [32] disengaging within the first 30 days of treatment.

The COVID-19 pandemic prompted a need to offer SUD treatment via telehealth [33,34,35], and concurrently yielded a 26% increase in alcohol use and a 16% increase in drug use [36]. Many states have revised regulatory policies, thereby creating the ability to provide outpatient SUD services via telehealth [37,38]. Existing research shows individuals are more likely to seek help online than in person [35], and that online options increase retention [39,40], drive patient satisfaction [41], and are as effective as in-person treatment [33,42,43,44].

Limited research has been conducted around the feasibility, effectiveness, and outcomes of online SUD treatment, though existing research shows several positive outcomes, including compliance with medication-assisted treatments [45]. Increased attendance and engagement have been demonstrated in virtual programs [46,47,48] and underscore the feasibility of such programming. Many individuals with higher severity SUDs require more intensive treatment inclusive of individual and group therapy over an extended period of time, otherwise known as Intensive Outpatient Programming (IOP). According to the American Society of Addiction Medicine [49], the IOP level of care must include 9–19 h of structured treatment per week delivered by professionals in the SUD field. The ASAM criteria assess individuals across six dimensions, which include the following:Intoxication, withdrawal, and addiction medications;Biomedical conditions;Psychiatric and cognitive conditions;Substance-use-related risks;Recovery environment interactions;Person-centered considerations.

The criteria for IOP eligibility are defined as follows:Minimal risk of severe withdrawal symptoms, manageable on an outpatient level;Medically stable, or not a distraction from treatment;May include the presence of mild psychiatric symptoms, which need monitoring and may distract from recovery;High likelihood of relapse without structured support and close monitoring several times per week;Recovery environment is not supportive, but with structure and support, individuals can cope.

IOP treatment is effective [50,51], and there remains a growing need for such levels of care in response to rising SUDs within the United States. Despite the need, there is limited research evaluating virtual SUD IOP treatment [33,34,44,48].

This preliminary study aims to add to the current literature around telehealth treatment for SUDs and address the gap that exists in the literature related to the feasibility of virtual SUD IOP treatment. As a result, the goal of the present study was to examine participation in individual therapy, group therapy, and asynchronous evidenced-based coursework.

## 2. Methods and Materials

### 2.1. Participants

Participants who were enrolled in the organization’s intensive outpatient SUD program between 2021 and 2023 were included in the study sample (*n* = 4725). De-identified data from the electronic health record of the organization were used for this study.

The sample (*n* = 4724) included adults aged 18+ (mean age = 42). Most identified as white (79.42%) and reported their socio-economic stability as adequate (75.42%). A total of 49.09% identified as male and 46.04% identified as female, with 4.87% not responding. Participants were geographically distributed across the United States (see Table 1).

To be eligible for IOP telehealth services, participants were required to be psychiatrically and medically stable, present with moderate to severe SUD symptoms meeting the ASAM criteria requiring an IOP level of care, have access to a private and secure area for sessions, and have reliable internet access. Those experiencing acute symptoms of withdrawal, active psychosis, medical issues impacting their ability to participate in group sessions, and/or active eating disorders with high-risk symptoms, and/or were in imminent risk of commiting suicide or homicide were excluded and provided with appropriate referral for care.

### 2.2. Procedures

The study was approved by the WCG Institutional Review Board and was given exempt status. Upon admission, all participants signed an admissions agreement, which included how their information and data may be used, their rights and responsibilities as a participant, and assurance of their digital rights.

Participants who attended SUD treatment received both group and individual therapy. The IOP program consisted of three weekly 3 h groups, one 50 min individual session per week, and digitally delivered asynchronous coursework that taught participants evidenced-based treatment concepts and skills. All sessions incorporated evidenced-based treatment modalities, such cognitive behavioral therapy (CBT), Dialectical Behavioral Therapy (DBT), and motivational interviewing (MI) skills. After completion of the IOP program, participants had the option to step down and continue outpatient care. Participants who did not complete the IOP program were provided with referrals to appropriate resources.

#### 2.2.1. IOP Program Model Details

All virtual IOP care was provided by licensed therapists with expertise in both SUD treatment and group-based treatment. Clinician competency was assessed through an extensive interview process to assess professional experience, clinical expertise in treating the SUD population, training, and skills, followed by a mock group in which therapists were observed facilitating a 40 min group to demonstrate proficiency. Clinicians then completed two weeks of training and met weekly with their supervisor and mentor on an ongoing basis. Training and supervision covered the following: evidence-based modalities and phase work utilized in the program, policies and procedures of the organization, engaging groups and individuals in the telehealth environment, and navigating the organization’s electronic medical record.

While participating in the IOP program, participants were expected to attend sessions consistently, with more than three absences triggering removal from the group until attendance was completed in an individual session to identify and ameliorate barriers to treatment. During this session, a recommitment plan was established and participants were informed of the risk of discharge from the IOP if consistent attendance and abstinence were not maintained. The IOP program is an abstinence-based program, though isolated relapse(s) were not immediate grounds for a referral to a higher level of care. Continued use/relapse, with consideration of ASAM criteria, triggered a referral to a higher level of care.

In order to complete the program successfully, participants were expected to complete the majority of their phase work and meet the goals identified on their individualized treatment plan. They were also expected to actively participate in groups and individual therapy sessions. Additionally, participants were expected to maintain ongoing abstinence, with at least 60 days of abstinence from all substances. The anticipated length of time to complete the program including asynchronous phase work assignments was approximately 4–5 months or less based on participant response to care.

#### 2.2.2. Phase Work

The asynchronous assignments were organized into five distinct phases, with five to eight assignments in each phase. Phase work follows evidenced-based approaches to SUD treatment including MI, CBT, and relapse prevention strategies. Throughout the program, participants were presented with concepts and educational content using these approaches, and completed relevant responses as part of their homework. Participants completed assignments, reviewed them in individual therapy sessions, and presented them in group sessions in order to obtain feedback.

Phase one focused on motivation for sobriety, phase two on CBT skills as applied to substance use disorders, phase three on developing a comprehensive relapse prevention plan, phase four on developing a sober support network, and phase five on creating a comprehensive discharge plan to aid in maintaining sobriety (Table 2).

### 2.3. Group Therapy

All intensive outpatient groups included 90 min of process and interactive support, and 90 min of psychoeducation on topics such as relapse prevention, coping skills, CBT and DBT skills, and social support. All group sessions were delivered with active therapist and participant participation using a HIPAA–compliant video conferencing platform.

### 2.4. Individual Therapy

Individual sessions included ongoing assessment of functioning based on ASAM criteria as well as participant-centered treatment techniques such as examination of goals set forth by the individual in the treatment plan developed collaboratively with their individual therapist at the outset of treatment. Progress in phase work assignment completion was monitored and reviewed prior to presenting it in group sessions. Drug and/or alcohol testing was completed as needed during individual sessions. All individual sessions were delivered using a HIPAA–compliant video conferencing platform.

### 2.5. Measures

The current study explored existing, de-identified data captured in the organization’s electronic health record. The organization used the Salesforce EHR for various functions, including clinical documentation and attendance, and was used to measure engagement among participants at 30, 60, 90, and 120 days of treatment. Engagement was defined as those actively participating in the IOP program at each designated interval (i.e., 30, 60, 90, and 120 days). Reasons for discontinuing treatment included the following: disengagement, discharged against clinical advice, referral to a higher level of care, and unknown/other when the reason for discharge was not available. Successful completion of the IOP was defined by a step down to an outpatient program, or discharge from the IOP after achievement of the identified treatment goals. Treatment goals were developed collaboratively with the participant, and all participants were aware that they were enrolling in an abstinence-based program.

Participant abstinence was tracked for participants who completed at least 30 days of treatment. Streaks of abstinence were defined as participants who maintained 30, 60, 90, 120, and/or over 120 days of abstinence at any given period of time while engaged in the program through their self-report.

Participant engagement factors, including continued attendance, participation in the program, and engagement with asynchronous assignments, were used to assess the feasibility of providing virtual SUD care.

### 2.6. Data Analysis

The feasibility of providing enhanced SUD IOP treatment via telehealth was evaluated using descriptive methods. Participant engagement and abstinence outcomes were evaluated at 30-, 60-, 90-, and 120-day intervals for all participants engaged in care. Final engagement outcomes were observed for all participants, resulting in six subgroups: completion of IOP and discharged, step down to outpatient care, disengagement, discharged against clinical advice, referred to a higher level of care, or unknown/other. Subgroups were analyzed for average length of stay in the program, and the percentage of phase work completed while in the program. For participants in the subgroup that stepped down to standard outpatient care, length of stay and phase work completion were inclusive of their participation in both IOP and standard outpatient care.

## 3. Results

### 3.1. Overall Engagement

A total of 4724 participants enrolled in the IOP program between 2021 and 2023. At 30 days of treatment, 79.86% (*n* = 3773) individuals remained engaged in treatment, 62.97% (*n* = 2975) at 60 days, 50.4% (*n* = 2381) at 90 days, and 39.91% (*n* = 1885) at 120 days. The mean length of treatment was 131 days (SD ± 139) for the total sample (*n* = 4724) (Figure 1).

### 3.2. Engagement: Successful Completion of IOP

A total of 402 (8.51%) participants successfully completed the IOP program and were discharged, while an additional 1692 (35.82%) participants successfully stepped down from the IOP to outpatient care. Together, these groups account for 44.33% of the total enrolled participants, and they demonstrated a successful response to care with no further IOP treatment needed (Table 3).

For those who completed the IOP and were discharged, the mean length of stay was 136.5 days, and a mean of 62.69% of phase work was completed. For participants who stepped down to outpatient care, the mean length of stay and percentage of phase work completed were 234.58 days and 73.31%, respectively, and were inclusive of both IOP and outpatient care (Table 3).

### 3.3. Engagement: Discontinued IOP

The most common reason for discontinuing the IOP was disengagement (23%, *n* = 1103). This group had a mean length of stay of 58.1 days (SD = ±54), with a mean of 5.87% (SD = ±12%) of phase work completed (Table 3). An additional 18.63% of participants (*n* = 880) were discharged against clinical advice, with a mean program length of 63 (SD = ±54) days, and 8.53% (SD = ±12.9%) of phase work completed (Table 3). A total of 10% of participants (*n* = 475) required referral to a higher level of care, attending a mean of 64.8 (SD = ±64) days of treatment with 8.53% (SD = ±14.6%) of phase work completed (Table 3). Finally, 3.6% of participants (*n* = 172) left the program for unknown reasons but attended a mean of 121.2 (SD = ±109) days of treatment and completed 31.07% (SD = ±35.4%) of individual phase work (Table 3).

### 3.4. Days of Consecutive Abstinence

Of the 3773 participants who remained engaged in care for at least 30 days, 91% self-reported at least 30 consecutive days of abstinence over the course of treatment. Of the 2976 participants who remained engaged in care for at least 60 days, 88% reported at least 60 consecutive days of abstinence, while 87% of the 2382 participants who engaged in care for at least 90 days reported abstinence for 90 consecutive days. A total of 87.01% of 1886 individuals engaged in care for at least 120 days reported at least 120 consecutive days of abstinence while in treatment (Table 4).

## 4. Discussion

This study investigated the feasibility of providing SUD IOP treatment via telehealth through an evaluation of participant engagement in care. The results provide support for feasible implementation of a nationwide virtual IOP with high rates of participant engagement and promising support for participants to remain abstinent from substances while in an IOP. Over 4700 participants were treated over the course of two years, with over 2000 participants successfully completing the IOP. The benefits of the IOP with respect to treatment engagement and sustained abstinence were broadly evident as nearly 80% of participants completed at least 30 days of care and maintained at least 30 days of consecutive abstinence during their treatment per their self-report. The high rates of participant engagement in the virtual IOP exceeded engagement estimates in conventional in-person IOP care (50–68% at 30 days [31,32]), supporting the enhanced benefit of a virtual care model for moderate to severe substance use conditions.

Nearly 45% of participants demonstrated a successful response to care and no longer required IOP treatment. This important finding underscores the value of telehealth options that can effectively stabilize more severe SUDs, and transition participants to outpatient care that requires less time, resources, and costs for participants, health systems, and payers [52]. By reducing barriers and providing care in a less restrictive setting, access to care can be increased to allow for sustained gains and ongoing progress in treatment.

Participants with SUDs often experience ambivalence related to their treatment and present with a high risk of return to use [53]. However, participants in the current study demonstrated encouraging rates of reported abstinence and treatment engagement. This may be explained by the quality of care provided through the IOP program, which included evidence-based techniques (CBT, DBT, and MI) across all components of treatment. These techniques were designed to address known barriers to engagement and treatment progress and were administered by trained clinicians with expertise in SUDs. As a result, participants engaged in effective treatment practices across individual therapy, group therapy, and asynchronous phase work that reinforced key concepts and skills continuously. That is, participants were able to receive therapeutic content in three specific settings to allow for enhanced acquisition of skills with the potential to identify and intervene in potential relapse scenarios.

Asynchronous phase work was added as an enhancement to routine IOP care and was implemented specifically for virtual administration. Participant engagement in individual phase work varied, with general trends suggesting that those who remained engaged in the IOP also completed their phase work. Those who finished the IOP also completed over 70% of their phase work, with phase work completion rates diminishing with reduced engagement in care. This finding is consistent with expectations that treatment targeting motivation and engagement leads to sustained progress and effort in treatment [52]. While the specific attributed effect of phase work in care outcomes cannot be observed in this current study, the results suggest that such additions to IOP do not hinder engagement, and likely support overall treatment. Additional research using randomized trials may be conducted to further investigate the role of asynchronous work in an IOP.

### 4.1. Limitations

This study was limited to a review of electronic medical records using archival data analysis of participant engagement and participation in treatment in an established IOP. A lack of accessibility of clinical outcome data limited the analytic design to descriptive analysis. As a result, follow-up evaluation of participants’ long-term recovery and additional clinical outcomes were not available. Nor were specific outcomes related to co-occurring mental health symptoms, and the reasons for disengagement in treatment could not always be ascertained. Additionally, provider and participant satisfaction and acceptability outcomes were not examined. Participant abstinence was limited to self-report, and relapse rates were unavailable, which would have provided further context to participants’ overall response to care. Notwithstanding, any sustained abstinence is a notable achievement that is predictive of recovery maintenance and progress [54,55,56]. The majority of participants presented with an alcohol use disorder, and other substance use diagnoses were underrepresented in the sample. Finally, there was a high degree of variability in the distribution of the length of stay and phase work completion outcomes across the sample, which should be accounted for when interpreting the results.

There are some notable considerations and challenges related with virtual IOP care delivery. Participants in an IOP may be more likely to experience high-risk events that require the mobilization of emergency services and immediate in-person evaluation. IOPs should proactively plan for such events with clear risk response protocols to address participant safety. In addition, there is the potential for fatigue in participating in video-based care for extended periods of time. Telehealth groups should include breaks, multi-modal content, and participation, and caution should be taken to ensure confidentiality for all group participants.

### 4.2. Future Directions of Research

Several areas of future research are needed to expand this work. First, additional studies exploring the clinical effectiveness of virtual IOP programs should be conducted using inferential statistics to evaluate clinical outcomes. Such studies would enhance the validity of these preliminary outcomes and provide further evidence to support the utility of virtual SUD IOP treatments. Further, capturing acceptability and satisfaction data from participants and providers would provide additional insights into the feasibility of virtual SUD IOP programming. In addition, rigorous designs such as a randomized controlled trial to compare the outcomes of in-person programs and virtual IOPs would help identify the unique benefits of virtual care.

Investigating reasons for participant engagement and disengagement as well as quality of engagement among individuals and groups, and with asynchronous treatment would help to understand future targets for program development. Given the high rates of co-occurring mental health disorders and SUDs, future studies would benefit from evaluating specific mental health symptoms and outcomes, and following key treatment outcomes longitudinally.

## 5. Conclusions

This study contributes to the literature by examining the feasibility of a virtual IOP for SUDs at scale by examining participant engagement factors. With few studies investigating telehealth IOP care models, this study provides preliminary insights that support the use of virtual care for participants with moderate to severe SUDs. The results indicate that a virtual IOP can be scaled nationwide effectively with high rates of participant engagement in individual therapy, group therapy, and asynchronous coursework supporting treatment adherence.

## Figures and Tables

**Figure 1 healthcare-12-02554-f001:**
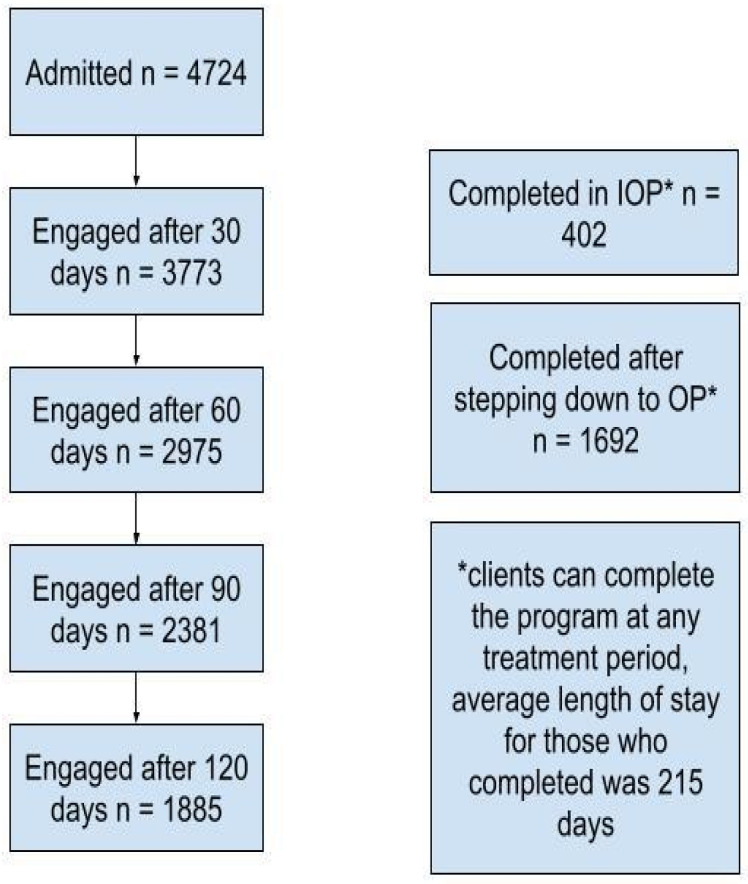
Engagement and disengagement in program.

**Table 1 healthcare-12-02554-t001:** Demographic information.

Characteristic	Study Sample (*n* = 4724)
Age	m = 42.3 (SD = ±11.4)
Sex	
Male	49.1%
Female	46.0%
No information	4.9%
Ethnicity	
White	79.4%
Hispanic/Latino	7.5%
Black or African American	6.7%
Some other race	2.2%
No information	2.1%
Asian	1.1%
American Indian or Alaska Native	0.8%
Native Hawaiian or Other Pacific Islander	0.2%
Socio-Economic Stability	
Adequate	75.4%
Above Average	13.8%
Poor	8.3%
Exceptional	1.5%
No information	0.9%
Educational Level	
College graduate	49.1%
Some college of techical school	30.6%
High school graduate	17.0%
Less than 12 years	2.5%
No information	0.8%
Sexual Orientation	
Heterosexual	90.14%
Homosexual	4.70%
Bisexual	3.49%
No information	1.63%
Transgendered	0.04%
Military	
No	92.8%
Yes	6.7%
No information	0.5%
Currently Employed	
Yes	80.1%
No	19.1%
No information	0.8%
Substance of Abuse	
Alcohol	78.11%
Amphetamine/Stimulant	6.82%
Opioid	5.25%
Cannabis	3.34%
Cocaine	3.20%
Other/Unknown Substance	1.91%
Sedative/Hypnotic/Anxiolytic	1.08%
Hallucinogen	0.23%
Inhalant	0.06%

**Table 2 healthcare-12-02554-t002:** Phase work assignments and description.

Phase	Concepts
Phase One Motivation to Change	Enhancing motivation and readiness change; use of motivational interviewing concepts
Phase Two Awareness	Developing awareness of thoughts, feelings, behaviors, and actions; use of CBT concepts
Phase Three Relapse Prevention	Developing a comprehensive relapse prevention plan; identifying and managing threats to recovery
Phase 4 Sober Support	Identifying and enhancing a sober support network
Phase 5 Discharge Planning	Creating a comprehensive discharge plan to maintain ongoing recovery

**Table 3 healthcare-12-02554-t003:** Reasons for disengagement.

Admitted to IOP Program (*n* = 4724)	*n* (%)	Length of Stay (Days)	% of Phase Work Completed
**Discontinued IOP**			
Disengaged	1103 (23.3%)	m = 58.1 (SD = ±54)	m = 5.9% (SD = ±12%)
Discharged against advice	880 (18.6%)	m = 63.0 (SD = ±54)	m = 9.0% (SD = ±12.9%)
Referred to a higher level of care	475 (10.1%)	m = 64.8 (SD = ±64)	m = 8.5% (SD = ±14.6%)
Unknown/other	172 (3.6%)	m = 121.2 (SD = ±109)	m = 31.1% (SD = ±35.4%)
**Successful Completion of IOP**			
Stepped down to outpatient	1692 (35.8%)	m = 234.6 * (SD = ±171)	m = 73.3% * (SD = ±36.0%)
Completed IOP and discharged	402 (8.5%)	m = 136.5 (SD = ±72)	m = 62.7% (SD = ±39.8%)

* includes treatment in outpatient.

**Table 4 healthcare-12-02554-t004:** Days of consecutive abstinence.

Total Sample	*n* = 4724
**30 days**	
Remained in treatment for at least 30 days:	*n* = 3773
Reported at least 30 days of abstinence while in treatment:	90.99%
**60 days**	
Remained in treatment for at least 60 days:	*n* = 2975
Reported at least 60 days of abstinence while in treatment:	87.87%
**90 days**	
Remained in treatment for at least 90 days:	*n* = 2381
Reported at least 90 days of abstinence while in treatment:	86.69%
**120 days**	
Remained in treatment for at least 120 days:	*n* = 1885
Reported at least 120 days of abstinence while in treatment:	87.00%

## Data Availability

Data is contained within the article.

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
