# Peer review of "Patient Engagement in Providing Telehealth SUD IOP Treatment: A Retrospective Cohort Study"

_healthcare, 2024, doi:10.3390/healthcare12242554_

Round 1

Reviewer 1 Report

Comments and Suggestions for Authors

The study highlights the growing public health challenge of SUD and the barriers that often prevent individuals from accessing care. It posits that telehealth interventions may offer a viable solution by improving access and engagement with treatment.

However, one major notable limitation of this study, is the absence of hypothesis testing or statistical analysis, which limits the ability to generalize or validate findings rigorously. While descriptive results provide an encouraging indication of telehealth IOP’s potential, without statistical tools or hypothesis testing, conclusions regarding its effectiveness remain preliminary. Incorporating quantitative analyses would strengthen the study’s validity and contribute more robust evidence to support the feasibility and efficacy claims.

Overall, while this study presents an innovative approach to overcoming access barriers for SUD treatment through telehealth, further research with rigorous statistical evaluation is essential for substantiating its findings.

Author Response

Please disregard attachment as line numbers were incorrect due to a formatting issue. please see below:

Comments 1: However, one major notable limitation of this study, is the absence of hypothesis testing or statistical analysis, which limits the ability to generalize or validate findings rigorously. While descriptive results provide an encouraging indication of telehealth IOP’s potential, without statistical tools or hypothesis testing, conclusions regarding its effectiveness remain preliminary. Incorporating quantitative analyses would strengthen the study’s validity and contribute more robust evidence to support the feasibility and efficacy claims.

Response 1: We have revised the introduction, discussion, conclusion, and limitations sections to emphasize this point and noted the preliminary nature of the paper, including removing reference to effectiveness of treatment and speaking only to feasibility. While we agree with this recommendation, inferential statistics were beyond the scope of this study due to the nature of available data. We have noted the need for more robust and rigorous statistical analysis in the future directions of research section and have further discussed the limitations of the data set in the limitations section.

Reviewer 2 Report

Comments and Suggestions for Authors

GENERAL COMMENTS

The number of participants at the start of the study (N=4724) is impressive, as is their gender balance. As a major flaw, the assessment of participant sobriety was based on their self reports. The attrition rate of 60%( i.e., 2839 of 4724, see fig1) was enormous. Considering this poor result, trumpeting and citing a misleading number at the beginng of the Discussion  : secion (lines 240ff) “the benefits of the IOP … were broadly evident as nearly 80% of participants completed at least 30-days of care …” is unwarranted. The authors must tone down this marketing statement considerably. The same goes for the Conclusion section (lines 307ff): “… meaningfully contributes …” and “… that support the use of virtual care for patients with moderate to severe SUDs …” are conclusions NOT supported by the authors’ results. The authors must tone down their overly optimistic conclusions and must emphasize that the assessment of sobriety was based solely on the participants’ self reports.

Also, the pharmacologic class of abused substances must be declared in the demographic information in table 1.

SPECIFIC ITEMS

line 16 and 234ff: “90.99% is unnecessary sophistication and pseudoscientific jargon. Please constrict to “91%). The same goes for “87.87%” and “86.69%” (these numbers read like a joke).

Author Response

Please disregard attachment as line numbers were incorrect due to a formatting issue. please see below:

Response to Reviewer 2 Comments

Point-by-point response to Comments and Suggestions for Authors

Comments 1: The attrition rate of 60%( i.e., 2839 of 4724, see fig1) was enormous. Considering this poor result, trumpeting and citing a misleading number at the beginning of the Discussion : secion (lines 240ff) “the benefits of the IOP … were broadly evident as nearly 80% of participants completed at least 30-days of care …” is unwarranted.

Response 1: We’d like to provide clarification to the above comment. In our paper, Figure 1 refers to engagement rates, not rate of attrition. At 60 days, 63% of the sample remained engaged in treatment, at 30 days, 80% remained engaged. We believe that this figure demonstrates notable rates of engagement that speak to the feasibility of implementing SUD IOP treatment via telehealth.

Comments 2: The same goes for the Conclusion section (lines 307ff): “… meaningfully contributes …” and “… that support the use of virtual care for patients with moderate to severe SUDs …” are conclusions NOT supported by the authors’ results. The authors must tone down their overly optimistic conclusions and must emphasize that the assessment of sobriety was based solely on the participants’ self reports.

Response 2: After careful review, we agree that the claims of effectiveness require additional data to support this conclusion. We have carefully revised the introduction, discussion, conclusion, and limitations sections to emphasize this point and noted the preliminary nature of the paper, including removing reference to effectiveness of treatment, focusing primarily on feasibility and engagement. We have also clarified reports of abstinence are based on self report, emphasizing the potential for lapses in abstinence to be unreported. Please see lines: 81, removal of second objective, removal of references to effectiveness of treatment (lines 192, 243, 285). We have also emphasized the limitation of self reports of abstinence and inability to verify this in the limitations section.

Comments 3: Also, the pharmacologic class of abused substances must be declared in the demographic information in table 1.

Response 3: This information has been added to the demographic information in table 1.

Comments 4: line 16 and 234ff: “90.99% is unnecessary sophistication and pseudoscientific jargon. Please constrict to “91%). The same goes for “87.87%” and “86.69%” (these numbers read like a joke)

Response 4: We have rounded these numbers up.

Reviewer 3 Report

Comments and Suggestions for Authors

Despite the good writing of the manuscript, I think the authors should make major changes before it is considered for publication. In its current version, the manuscript has weaknesses that are of great concern to me and that I believe should be addressed. I indicate below which aspects should be improved in my opinion.

1. The subsection 2.3. IOP Program Model Details, is actually part of the procedure section and should not be a separate method section. The same is true for section 2.4. Phase Work, both could be numbered as subsections 2.2.1. and 2.2.2. within the procedure section.

2. I miss at least a basic introductory description of the characteristics of the electronic medical record in the measures section.

3. In the measures section, I suggest clarifying whether therapeutic goals are mandatory and set by the agency or whether patients can make the decision to reduce and not be totally abstinent as a goal.

4. The analysis section is very weak, not providing an adequate description of the types of analyses performed (e.g., univariate analysis of percentages, mean values, etc.). For an adequate assessment of sobriety (I would better use the term abstinence) for each time interval the authors should have used bivariate analysis or inferential statistics.

5. I have not seen any section that refers to approval of the study by ethics committees, administration of informed consent, assurance of digital rights, or following the Helsinki declaration.

6. In section 3.2 of the results, the authors should provide in the text the corresponding dispersion statistic (e.g., standard deviation) each time they provide mean or central tendency values.

7. The authors make a very confusing definition of symptom reduction, defining it as days in abstinence, which is really not an operational and precise definition that can detract from the contribution of the manuscript. I suggest that the authors define abstinence as a treatment outcome and avoid the current fuzzy definition of symptom reduction. Furthermore, in the limitations of the study the authors note that they do not evaluate clinical outcomes, when in fact abstinence is considered a clinical outcome.

8. The authors point out in the discussion, especially in their conclusions, that the study contributes significantly to the literature by examining feasibility and effectiveness. This is an incorrect conclusion, and in any case they provide information on feasibility by means of mean values and percentages without inferential statistical comparison, it is merely descriptive and therefore does not make any significant contribution to the effectiveness of the treatment. No statistical comparisons of groups are made.

I hope these suggestions contribute to enhancing the quality of the work.

Author Response

Please disregard attachment as line numbers were incorrect due to a formatting issue. please see below:

Response to Reviewer 3 Comments

3. Point-by-point response to Comments and Suggestions for Authors

Comments 1: The subsection 2.3. IOP Program Model Details, is actually part of the procedure section and should not be a separate method section. The same is true for section 2.4. Phase Work, both could be numbered as subsections 2.2.1. and 2.2.2. within the procedure section

Response 1: Thank you for this recommendation. We have edited subsections (lines 120 & 147) as suggested.

Comments 2: I miss at least a basic introductory description of the characteristics of the electronic medical record in the measures section.

Response 2: We agree and have added a brief description in the measures section, starting on line 175.

Comments 3: In the measures section, I suggest clarifying whether therapeutic goals are mandatory and set by the agency or whether

patients can make the decision to reduce and not be totally abstinent as a goal.

Response 3: Agree. We have clarified this starting on line 184 and updated text to include “Treatment goals were developed collaboratively with the participant and all participants were aware that they were enrolling in an abstinence based program.

Comments 4: The analysis section is very weak, not providing an adequate description of the types of analyses performed (e.g., univariate analysis of percentages, mean values, etc.). For an adequate assessment of sobriety (I would better use the term abstinence) for each time interval the authors should have used bivariate analysis or inferential statistics.

Response 4: We have revised the manuscript to use the term abstinence throughout. We have also, accordingly, revised the introduction, discussion, conclusion, and limitations sections to emphasize this point and noted the preliminary nature of the paper, including removing reference to effectiveness of treatment and speaking to feasibility. We have also clarified reports of abstinence are based on self report, emphasizing the potential for lapses in abstinence to be unreported. Pleas`e see lines: 81, removal of second objective after line removal of references to effectiveness of treatment (lines 192, 243). We have also emphasized the limitation of self reports of abstinence and inability to verify this in the limitations section.

Comments 5: I have not seen any section that refers to approval of the study by ethics committees, administration of informed consent, assurance of digital ights, or following the Helsinki declaration.

Response 5: We have expanded on the approval of the study by an ethics board in the procedures section (beginning on line 107) and it now reads: “The study was approved by the WCB Institutional Review Board and was given exempt status. Upon admissions, all participants signed an admissions agreement which included how their information and data may be used, their rights and responsibilities as a participant, and assurance of their digital rights.”

Comments 6: . In section 3.2 of the results, the authors should provide in the text the corresponding dispersion statistic (e.g., standard deviation) each time they provide mean or central tendency values.

Response 6: Standard deviations of means can be found in table 3 (line 232) and have been added in the text (line 207).

Comments 7: The authors make a very confusing definition of symptom reduction, defining it as days in abstinence, which is really not an operational and precise definition that can detract from the contribution of the manuscript. I suggest that the authors define abstinence as a treatment outcome and avoid the current fuzzy definition of symptom reduction. Furthermore, in the limitations of the study the authors note that they do not evaluate clinical outcomes, when in fact abstinence is considered a clinical outcome.

Response 7: Thank you for this feedback. We have removed ‘symptom reduction’ and clarified indicating days of abstinence as an outcome. (see line 233). We have noted in the limitations section that the measure of abstinence is based on self report, and have offered suggestions for future research to target additional clinical outcomes.

Comments 8: The authors point out in the discussion, especially in their conclusions, that the study contributes significantly to the literature by examining feasibility and effectiveness. This is an incorrect conclusion, and in any case they provide information on feasibility by means of mean values and percentages without inferential statistical comparison, it is merely descriptive and therefore does not make any significant contribution to the effectiveness of the treatment. No statistical comparisons of groups are made

Response 8: We agree and have made changes in the discussion, conclusion, and limitations sections, removing reference to treatment effectiveness and only speaking to the feasibility of the model with regard to treatment engagement.

Round 2

Reviewer 2 Report

Comments and Suggestions for Authors

acceptable

Author Response

Thank you for taking the time to review the revisions made.

Reviewer 3 Report

Comments and Suggestions for Authors

I congratulate the authors for the substantial improvements they have made to the manuscript based on the reviewers' suggestions. I do not have any new suggestions, I consider that the manuscript has improved enough to be considered suitable for publication.

Kind regards.

Author Response

Thank you for your response and taking the time to review our revisions.